# Structure of Staphylococcal Enterotoxin N: Implications for Binding Properties to Its Cellular Proteins

**DOI:** 10.3390/ijms20235921

**Published:** 2019-11-25

**Authors:** Chi Zeng, Zhaoxin Liu, Zhenggang Han

**Affiliations:** 1College of Biology and Pharmaceutical Engineering, Wuhan Polytechnic University, Wuhan 430023, China; czeng@whpu.edu.cn (C.Z.); jiangxinklzx@163.com (Z.L.); 2Hubei Province Fresh Food Engineering Research Center, Wuhan Polytechnic University, Wuhan 430023, China; 3Key Laboratory of Prevention and Control Agents for Animal Bacteriosis (Ministry of Agriculture), Institute of Animal Husbandry and Veterinary, Hubei Academy of Agricultural Sciences, Wuhan 430064, China; 4College of Life Sciences, Wuhan University, Wuhan 430072, China

**Keywords:** superantigens, staphylococcal enterotoxin, major histocompatibility complex class II, T-cell receptor

## Abstract

*Staphylococcus aureus* strains produce a unique family of immunostimulatory exotoxins termed as bacterial superantigens (SAgs), which cross-link major histocompatibility complex class II (MHC II) molecule and T-cell receptor (TCR) to stimulate large numbers of T cells at extremely low concentrations. SAgs are associated with food poisoning and toxic shock syndrome. To date, 26 genetically distinct staphylococcal SAgs have been reported. This study reports the first X-ray structure of newly characterized staphylococcal enterotoxin N (SEN). SEN possesses the classical two domain architecture that includes an N-terminal oligonucleotide-binding fold and a C-terminal β-grasp domain. Amino acid and structure alignments revealed that several critical amino acids that are proposed to be responsible for MHC II and TCR molecule engagements are variable in SEN, suggesting that SEN may adopt a different binding mode to its cellular receptors. This work helps better understand the mechanisms of action of SAgs.

## 1. Introduction

Superantigens (SAgs) are the critical virulence factors of *Staphylococcus aureus* [1,2]. SAgs can cross-bridge the major histocompatibility complex class II (MHC II) with T cell receptor (TCR) molecules to stimulate a massive proliferation of T cells (~20–30%) in a relatively nonspecific manner [3]. The dysregulated immune system leads to severe human illnesses such as toxic shock syndrome (TSS) [2,4,5].

SAgs are a group of simple, non-glycosylated proteins that are highly resistant to heat denaturation and proteolysis [6]. Mature SAgs have a relatively low molecular weight of 19,000–30,000 Da (220–240 amino acids) [2,3]. To date, more than 26 genetically distinct staphylococcal SAgs have been described, including TSS toxin-1 (TSST-1), staphylococcal enterotoxins (SEs), and SE-like (SE-l) SAgs [5,6,7,8]. SE-type superantigens are defined by their ability to induce emesis, and SE-l-type either lack of emetic activity or have not been formally investigated to induce emesis [9]. Staphylococcal SAgs are categorized based on their amino acid sequence divergence into four evolutionary groups: I, II, III, and V (SAg group IV is streptococcal exotoxins).

Although the characterized SAgs exhibit substantial sequence variability, they share a similar three-dimensional architecture, as demonstrated by the crystal structures of 11 SAgs [10]. The multiple X-ray structures of staphylococcal SAgs revealed that the protein is composed of an N-terminal oligonucleotide-binding (OB) fold and a C-terminal β-grasp domain [3,10]. Crystallographic and mutational studies have also provided considerable information with regard to the molecular interactions between SAgs and host targets for several SAgs, such as SEA, SEB, and SEH [9]. The identified binding regions on SAgs include the binding site for β-chain T cell receptor (TRBV) (in a shallow groove between two domains of SAgs), generic low-affinity MHC II (α-chain) binding site (in the OB-fold of SAgs), high-affinity MHC II (β-chain) binding site (in the β-grasp domain in many SAgs), and epithelial/endothelial/CD40/CD28 binding site [9]. According to the presence of certain receptor-binding sites, distinct SAgs exhibit variable frameworks with regard to SAgs/TCR/MHC complexes [9]. SAgs from groups I, III, and IV tricomplexes appear as beads on a string, whereas group II SAgs appear as a wedge between the TCR and MHC molecules [4,9]. A number of conserved amino acid residues in SAgs are involved in each of these binding regions. However, amino acid variations in individual SAgs are present by which different subsets of immune molecules are selected. For example, each SAg recognizes a relatively unique set of Vβ domains on TCRs, thereby, inducing massive activation of certain T-cell repertoire [5].

Except for the most common genetic types of SAgs (TSST-1, SEA, SEB, SEE, and SEH), most new types of SAgs characterized genetically and antigenically have not been studied in depth [3]. For example, SEL, SEM, SEN, and SEO, are frequently detected in strains isolated from patients with food poisoning [11,12,13,14,15]. However, their crystal structures and structural insights into interactions with host receptors have not been elucidated. The present study reports the X-ray structure of SEN. The SEN structure shows the classical two-domain superantigen architecture and is most similar to SEA. Along with amino acids and three-dimensional structure comparisons, several interesting amino acids and loop region variations in SEN were observed. The characteristics of SEN provide structural insights into the interactions of SEN with host cell proteins.

## 2. Results and Discussion

### 2.1. Overall Structure of SEN

Recombinant SEN production and purification were evaluated by sodium dodecyl sulfate-polyacrylamide gel electrophoresis (SDS-PAGE), which are summarized in Appendix A. The structure of SEN was solved by molecular replacement, and the final model was refined at 1.8 Å resolution (R_work_ = 0.171, R_free_ = 0.201, Table 1). There is one monomer per asymmetric unit. Overall, SEN is a compact molecule and shows the classic two-domain architecture of SAg proteins. The N-terminal OB-fold domain (residues 52–137) consists of five-stranded β-barrel (β1, β2, β3, β4, β5a, and β5b) and two 3_10_ helices (3_10_ 1 and 3_10_ 2). The C-terminal β-grasp domain of SEN consists of anti-paralleled β-strands (β6–β12) and α-helices (α4 and α5) (Figure 1). The β-grasp domain contains both the amino and carboxyl termini of the protein. Residues 27–51 from the N-terminal of SEN encompass one side of the β-grasp domain with α2 in this region situated in the middle of the two domains. α2, α4, and α5 form an approximate triangle.

SEN belongs to the group III SAg as revealed by multiple amino acid sequence alignment (Figure 2) [16]. SEN is structurally similar to those crystal structures of SEs, with a pairwise α-carbon root-mean-square deviation (RMSD) between 1.13 and 2.51 Å (Appendix A). SEN is most similar to the group III SAgs SEA (Protein Data Bank (PDB) entry: 1SXT), with an α-carbon RMSD of 1.13 Å. SEN is less similar to other group III SAgs SEE (PDB entry: 5FKA) and SEH (PDB entry: 2XN9), with an α-carbon RMSD of 1.23 and 1.38 Å, respectively. SEN is distant from group I SAgs SEI-X (PDB entry: 5U75) and TSST-1 (PDB entry: 2QIL), with an RMSD of 2.51 and 2.22 Å, respectively.

### 2.2. The TCR Binding Site of SEN

Crystal structures of SAgs in complex with TCRs (SEA-hTRBV7-9, PDB entry: 5FK9; SEB-hTRBV19, PDB entry: 4C56; SEC3-mTRBV13-2, PDB entry: 3BYT; SEE-hTRBV7-9, PDB entry: 5FKA; SEH-hTRAV27, PDB entry: 2XNA) reveal that TRBV (SEH is an exception, it is the only Vα-specific Sag that is known) engages SAgs at a shallow groove between the OB and β-grasp domains [4,17]. Several conserved hot-spot regions are responsible for SAg-TCR interactions [17,18]. Crucial amino acid residues that contribute to the complex formation common to all the five SAg-TCR complexes lie in two conserved regions: a strictly conserved asparagine on α2 and an aromatic residue pair (not strictly conserved, normally have at least one aromatic residue) on the β4–β5a loop [17,18,19]. These two regions contribute to the complex formation through hydrogen bonds and hydrophobic interactiosn, respectively [17,18]. Superposition of SEN and other SAg structures revealed that in the SEN structure, an asparagine (Asn44) is situated in a nearly identical position to that in the other SAgs (Figure 2 and Figure 3). However, the aromatic pair is not present in the equivalent position in SEN. The corresponding amino acids in the position are Gly113 and Asn114 (Figure 2 and Figure 3). These amino acid variations might confer TRBV specificity to SEN. Previous studies have shown that SEN can activate four TRBV subgroups (7, 8, 9, and 17), and it is only group III SAg that is able to activate the TRBV subgroup 17 [5].

A structural model of the SEN-hTRBV7-9 complex was obtained by superposition of SEN and SEA-hTRBV7-9 and performing energy minimization (Appendix A). The interaction interface between SAg and TRBV was calculated using the PDBePISA server [20]. The modeled SEN-hTRBV7-9 complex has an interface area of 865.0 A^2^, which is smaller than those of SEA-hTRBV7-9 (910.7 A^2^) and SEE-hTRBV7-9 (946.3 A^2^) structures. The intermolecular hydrogen bonds in SEN-hTRBV7-9 (3) are far less than those in SEA-hTRBV7-9 (12) and SEE-hTRBV7-9 (20) (Appendix A). Both amide nitrogen atoms of Gly113 and Asn114 in SEN form hydrogen bonds with Glu53 OE1 on the CDR2 loop of hTRBV7-9. Such a hydrogen bond network is present in SEA-hTRBV7-9 but not in SEE-hTRBV7-9. The strictly conserved asparagine (Asn44) in the SEN dose is not involved in the intermolecular hydrogen bond formation which is different from that in SEA-hTRBV7-9 and SEE-hTRBV7-9 (Appendix A).

### 2.3. MHC II-Binding Sites on SEN

Binding of SAgs to MHC class II molecules on antigen-presenting cells occurs prior to the engagement of TCRs. Two MHC II-binding sites are identified on SAgs. A generic low-affinity site located within the OB domains of all SAgs interacts with MHC II α-chain (MHCα). Another zinc-dependent, high-affinity site, within the β-grasp domains of SAgs from groups III, IV, and V, binds to MHC II β-chain (MHCβ) [4,5,9]. As a group III SAg, SEN has both MHC-binding regions. According to the pioneering crystallographic studies of SAg-MHC complexes (SEA-MHC, PDB entry: 1LO5; SEB-MHC, PDB entry: 4C56), a stretch of hydrophobic amino acids (Phe/Leu/His in SEA and Phe/Leu/Phe in SEB) on the β1–β2 loop constituting a hydrophobic ridge protrudes into a hydrophobic pocket in MHCα forming the main interaction between SAg and MHCα [21,22]. The structurally equivalent amino acid residues in SEN are Leu67/Leu68/Asn70, which are different from those in SEA and SEB (Figure 2 and Figure 4A,B). Other interactions mainly occur through a number of hydrogen bonds. However, the hydrogen-bond interactions are distinct for SEB and SEA. The number of hydrogen bonds between SEB and MHCα (11) is more than that observed in the SEA-MHCα complex (2) [21,22]. Therefore, amino acids involved in the hydrogen bond formation at the low-affinity binding site of SEN are difficult to predict. A structural model of SEN-MHCα was generated by energy minimization. The interface area of SEN-MHCα, SEA-MHCα, and SEB-MHCα are 541.7 A^2^, 560.8 A^2^, and 731.8 A^2^, respectively, which suggests that the mode for SEN to engage MHCα is similar to that of SEA but distinct to that of SEB. The small contact region has been proposed to lead to SEA binding 40-fold weaker to MHCα compare to that of SEB [21].

The interaction at the high-affinity site of SAgs is dependent on a zinc ion and displays a nanomolar range binding affinity [4]. As a group III SAg, SEN has a putative high-affinity MHCβ binding site. The crystal structure of SEH-MHCβ complex shows that the high-affinity site is located on the surface of the twisted β-sheet (β6, β9, β10, and β12). A histidine and an aspartate from β12 are the two protein ligands for zinc ion. The side chains of these two zinc-coordinating amino acids are stabilized through hydrogen bonds offered by serine and an asparagine from β6 and β12, respectively [23]. The ion-coordinating amino acids and their stabilizing amino acids are present in the structure of SEN (Figure 2 and Figure 4C,D). However, in the SEN structure, the amino acid stabilizing ion-coordinating histidine is replaced by asparagine (Asn205) from the β9–β10 loop (Figure 4C,D). A structural model of SEN-MHCβ was generated by energy minimization. The interface area of SEN-MHCβ (431.6 A^2^) is similar to that of SEH-MHCβ (439.1 A^2^), which suggests that SEN and SHE adopt a similar binding mode to the MHCβ molecule.

### 2.4. Other Functional Regions on SEN

In the C-terminal domain, a motif consisting of the β7–β8 loop, β8, and β8–a4 loops on SEB is the proposed binding site for CD28 molecule [24]. The amino acid constitution in the region is highly basic. The amino acid alignment indicated that this region is largely conserved among SAgs from different evolutionary groups (Figure 2). The conformation of the putative CD28 binding region in SEN is consistent with those in SEB (group II), SEH (group III), and SEI (group V), except for group I SAg, such as TSST-1 (Figure 5, lower). This functional region in SEN is more basic than others, as demonstrated by its four lysine residues (Figure 2).

SAgs from groups II and III have a characteristic cystine loop (on β4–β5a loop) at their N-terminal domain [4]. This loop is essential for the emetic activity of SAgs [4,25]. In the structure of SEN, cystine pair, Cys116 (on β4-β5a) and Cys126 (on β5a), forms a disulfide bond, which is in agreement with the observation in structures of SEB (PDB entry: 3SEB) and SEH (PDB entry: 2XN9) (Figure 5). The cystine loops in the structures of SEA (PDB entry: 1SXT) and SEE (PDB entry: 5FKA) are incomplete. The length and conformation of the cystine loop in SEN are comparable with that of SEH but much shorter than that of group II SAg, SEB (Figure 5). The presence of the cystine loop in SEN is consistent with its weakly emetic activity observed in primates [5,13].

## 3. Materials and Methods

### 3.1. Recombinant SEN Production

The nucleotide sequence of the *sen* gene is available in GenBank (accession number: AF285760). DNA fragments encoding residues 25-251 (removing the signal peptide: residues 1-24) were amplified from the genome of *S. aureus strain* (ATCC 29213) by using the primer pair: 5′-CGCGGATCCGAGAATCTGTACTTCCAGGGCGATGTAGACAAAAATGATT TA-3′ and 5′-ATAAGAATGCGGCCGCTTAATCTTTATATAAAAATACATCG-3′ (the underlined bases correspond to the restriction sites). The purified DNA fragments were cloned into the *Bam*HI/*Not*I sites of the expression vector pGEX-6P-1 (GE Healthcare Life Sciences, Beijing, China). Recombinant SEN was produced using *Escherichia coli* BL21 (DE3) cells. Overnight cultures of *E. coli* transformants were inoculated into 1 L of Luria–Bertani (LB) liquid medium containing 100 μg/mL ampicillin and grown at 37 °C with shaking (200 rpm). Isopropyl β-D-1-thiogalactopyranoside (final concentration of 0.2 mM) was added to the culture when the optical density at 600 nm reached ~0.6. After induction, the culture was incubated at 16 °C with shaking (180 rpm) for 16 h. The cells were harvested by centrifugation at 4500× *g* for 20 min at 4 °C, and the pellet was stored at −80 °C.

### 3.2. Recombinant SEN Purification

The cell pellet was resuspended in lysis buffer (10 mM Tris-HCl pH 8.0, 200 mM NaCl, 5% *v*/*v* glycerol, 0.3% *v*/*v* Triton X-100, 1 mM dithiothreitol, 0.1 mM phenylmethylsulfonyl fluoride) and broken via French pressure cell press. The cell lysate was centrifuged at 12,000× *g* for 30 min at 4 °C to remove insoluble debris. The supernatant was filtered through a 0.45 μm filter and passed through a glutathione S-transferase (GST) Sepharose affinity column (GE Healthcare Life Sciences, Beijing, China) equilibrated with 10 mM Tris-HCl pH 8.0. The target protein was eluted with 10 mM Tris-HCl pH 8.0 containing 10 mM reduced glutathione. The purified fusion protein was digested overnight at 4 °C with TEV protease (Sangon Biotech, Shanghai, China) and then passed through a GST Sepharose affinity column to remove the GST tag. The recombinant protein was then loaded onto a MonoQ 5/50 GL anion exchange column (GE Healthcare Life Sciences, Beijing, China) and eluted using a linear NaCl gradient. The last step of purification was conducted using a Superdex 200 10/300 GL column (GE Healthcare Life Sciences, Beijing, China) on ÄKTAprime plus system (GE Healthcare Life Sciences, Beijing, China). The gel filtration fractions were checked by SDS-PAGE, and pure fractions were pooled and concentrated to 10 mg/mL for crystallization experiments.

### 3.3. SEN Crystallization

Screening of crystallization condition was performed at 4 °C by using an automated liquid-handling robotic system (Formulatrix, Nano Transfer 8) and crystallization screening kits from Hampton Research (Aliso Viejo, CA, USA) (Crystal Screen Suite, Crystal Screen 2 Suite, PEG/Ion Screen Suite, PEG/Ion 2 Screen Suite, Index Suite, PEGRx 1 Suite, and PEGRx 2 Suite) and Molecular Dimensions (MIDAS Suite, SG1 Screen Suite, PACT premier Suite, Structure Screen 1 Suite, Structure Screen 2 Suite, Morpheus Suite, Stura Footprint Screen Suite, 3D Structure Screen Suite, MultiXtal Suite, MacroSol Suite, JCSG-plus Suite, and ProPlex Suite). Sitting drops (0.5 μL) consisting of 0.25 μL of the protein solution and 0.25 μL of the reservoir solution were equilibrated against 30 μL of the reservoir solution in 96-well MRC plates (Molecular Dimensions, Newmarket, Suffolk, UK). After three days, tiny polycrystals were grown under the condition containing 2% (*v*/*v*) Tacsimate™ pH 4.0 and 16% (*w*/*v*) polyethylene glycol (PEG) 3350. Diffraction quality crystals were obtained under the optimized conditions containing 2% (*v*/*v*) Tacsimate™ pH 4.0 and 12% (*w*/*v*) PEG 3350.

### 3.4. Data Collection, Processing, and Structure Determination

Crystals used for diffraction data collection were soaked in cryoprotectant (mother liquor containing 20% glycerol) for 10 s before flash cooling in a liquid nitrogen stream. Diffraction data were collected at 100 K on beamline BL17U1 (wavelength of 0.97915 Å) (Shanghai Synchrotron Radiation Facility (SSRF), Shanghai, China). The data were indexed using iMosflm [26] and scaled using SCALA [27]. The structure was determined by molecular replacement with the structure of SEA (PDB code: 1LO5) [21] as the search model. Model rebuilding and refinement were carried out using Coot [28] and REFMAC5 [29], respectively. During refinement, 5% of the reflections were set aside for the calculation of the free R-factor [30]. Table 1 shows a summary of the diffraction data collection, processing, and refinement. The atomic coordinate and structure factors have been deposited in the PDB under accession entry 6KRY. Pairwise RMSD values between SEN and other SAg structures were calculated using SuperPose (http://superpose.wishartlab.com/). Structure-based sequence alignment was performed on the T-Coffee server [31] and ESPript3.0 [32]. Protein interfaces were calculated by PDBePISA server [20]. Structural figures were prepared using PyMOL (Schrödinger).

## 4. Conclusions

In this study, the structure of staphylococcal enterotoxin SEN was determined by X-ray crystallography. The structure of SEN displays the conserved N-terminal OB-fold and C-terminal β-grasp domain. Deep structural analysis revealed that there are a number of interesting divergences at the functional regions of SEN. The notable difference is the TRBV interface on SEN, which is not conserved in comparison to that on the other reported SAgs. The conserved aromatic pair involved in TRBV binding does not exist in SEN. The MHCα binding site (low-affinity site) also contains a certain degree of variation, in particular, those on the crucial hydrophobic ridge. These structural divergences will alert the binding mode (critical amino acids involved and binding affinity) of SEN with its cellular receptors. Our study helps better understand the mechanisms of action of SAgs and also implies that the development of the SAg blocker with broad-spectrum activity still is a challenge.

## Figures and Tables

**Figure 1 ijms-20-05921-f001:**
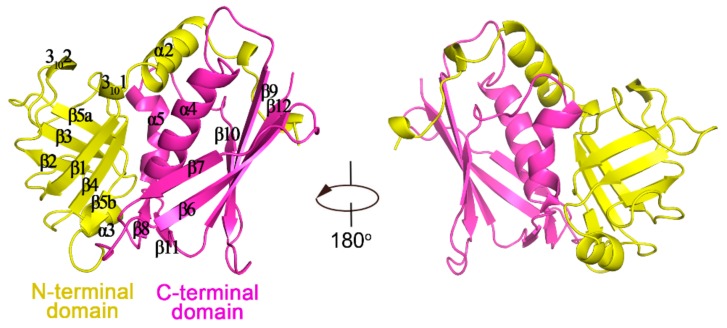
Crystal structure of staphylococcal enterotoxin N (SEN). The structure is represented with N-terminal and C-terminal domains shown in yellow and magenta, respectively. Secondary structural elements are labeled on the structure.

**Figure 2 ijms-20-05921-f002:**
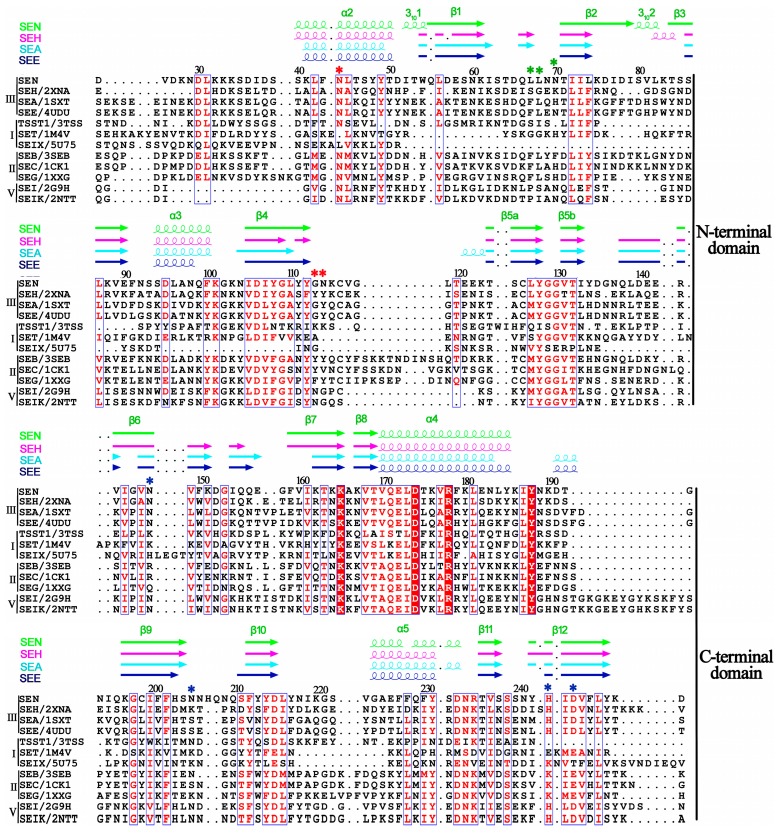
Amino acid sequence alignment of superantigens (SAgs) from different evolutionary groups. The secondary structures of group III SAgs (SEA, SEE, SEH, and SEN) are shown above the sequences. The N-terminal parts of the sequences are not shown. The amino acid residue numbering is according to SEN. Amino acid residues involved in the T-cell receptor binding site, low-affinity major histocompatibility complex (MHC) class II binding site and high-affinity MHC binding site are highlighted by red, green, and blue asterisks, respectively.

**Figure 3 ijms-20-05921-f003:**
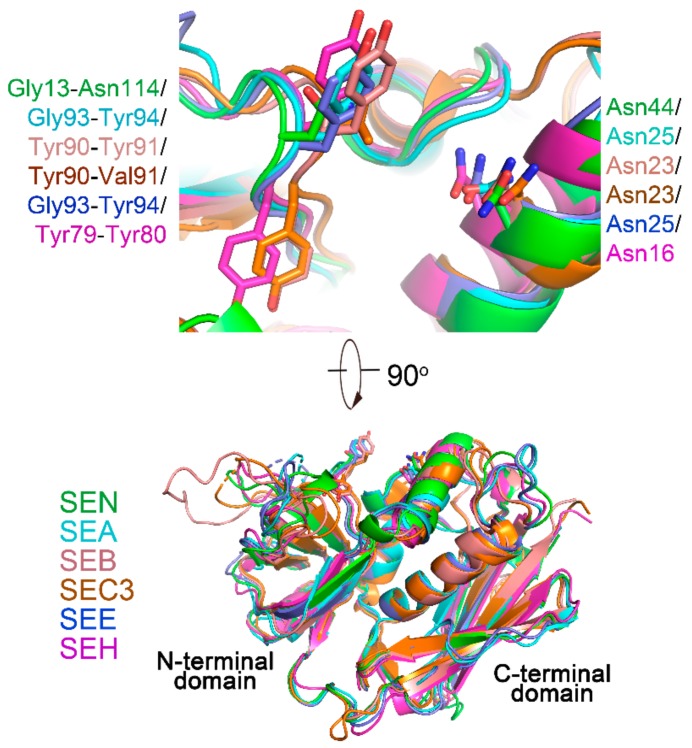
TCR binding site on SEN. Superpositions of SAgs from crystal structures of SAg-TCR complexes (SEA-hTRBV7-9, PDB entry: 5FK9; SEB-hTRBV19, PDB entry: 4C56; SEC3-mTRBV13-2, PDB entry: 3BYT; SEE-hTRBV7-9, PDB entry: 5FKA; SEH-hTRAV27, PDB entry: 2XNA) (**lower**) and close-up view of the T-cell receptor binding site (**upper**).

**Figure 4 ijms-20-05921-f004:**
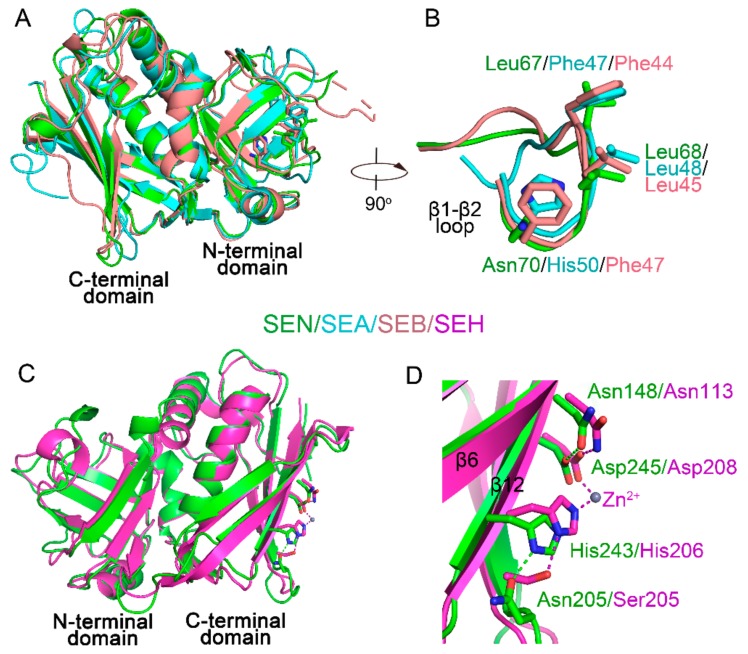
MHC II-binding sites on SEN. (**A**) Structural alignment of SEN, SEA (PDB entry: 1LO5), and SEB (PDB entry: 4C56). (**B**) Close-up view of the low-affinity MHC II-binding sites on SEN. (**C**) Structural alignment of SEN and SEH (PDB entry: 1HXY). (**D**) Close-up view of the high-affinity MHC II-binding sites on SEN.

**Figure 5 ijms-20-05921-f005:**
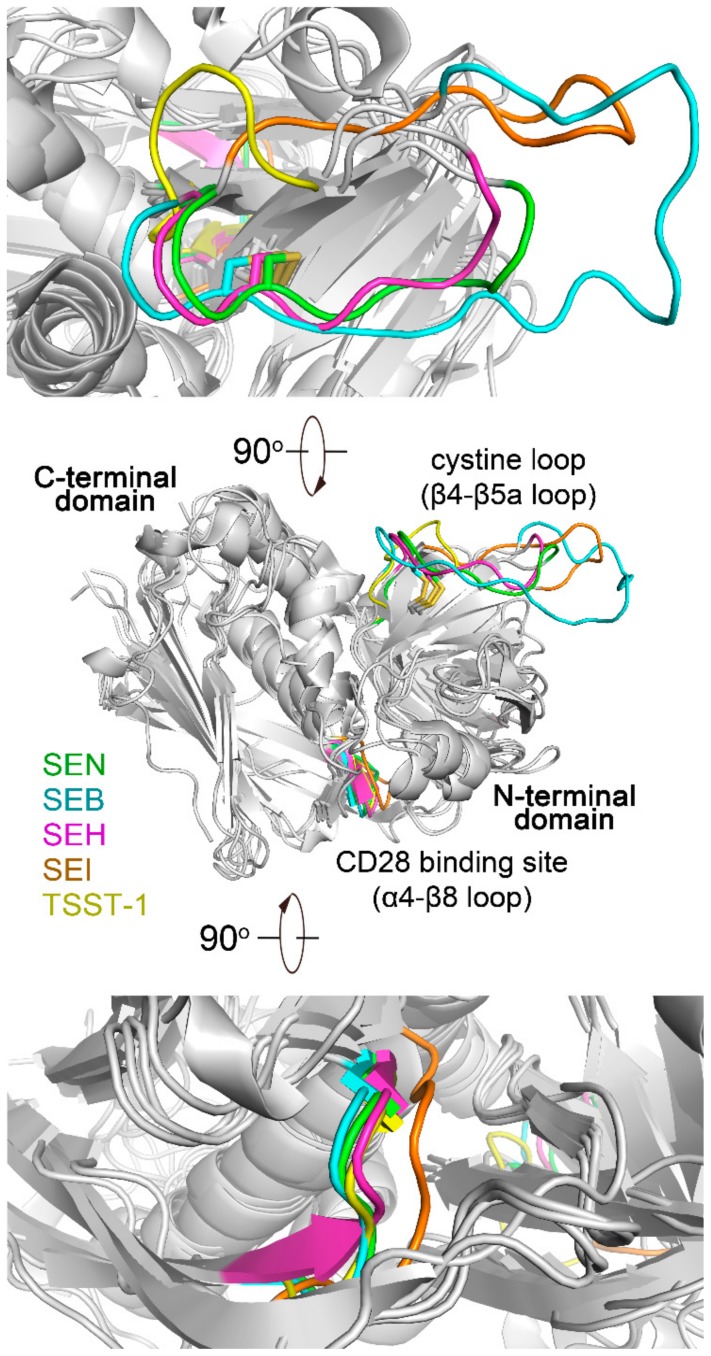
CD28 binding sites and cystine loops on SEN and the SAgs from different evolutionary groups. Crystal structures of SEN, SEB (PDB entry: 3SEB, group II), SEH (PDB entry: 2XN9, group III), SEI (PDB entry: 2G9H, group V), and TSST-1 (PDB entry: 2QI1, group I) are superimposed. The CD28 binding regions and cystine loops are presented in color, and the other regions of the protein are shown in gray. CD28 binding sites and the cystine loops are located on the C-terminal domain of SAgs. SEI and TSST-1 lack a cystine pair on the β4–β5a loop. Close-up view of the CD28 binding sites and the cystine loops are shown lower and upper parts, respectively.

**Table 1 ijms-20-05921-t001:** Data collection and refinement statistics. Values in parentheses are for the highest resolution shell.

Data Collection	
X-ray source	BL17U1, SSRF
Wavelength, Å	0.97915
Space group	P12_1_1
Unit cell parameters, Å, deg.	a = 42.00, b = 64.42, c = 50.14, β = 110.03
Resolution range, Å	64.42–1.80 (1.90–1.80)
No. reflections	137,766 (19,605)
Unique reflections	23007 (3343)
R_merge_	0.113 (0.580)
CC_1/2_	0.996 (0.859)
I/σ(I)	11.4 (3.5)
Redundancy	6.0 (5.9)
Completeness, %	98.5 (98.4)
Wilson B factor, Å^2^	11.5
**Refinement statistics**	
Refinement resolution range, Å	47.11–1.80
No. reflections	
*R*_work_/*R*_free_	0.171/0.208
No. atoms	
Protein	1821
Water	171
Average B factor, Å^2^	19.0
R.m.s. deviations	
Bond lengths, Å	0.0113
Bond angles, deg.	1.5415
Ramachandran statistics	
Favored, %	99
Allowed, %	1
Outliers, %	0

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
