# Peer review of "Structure of Staphylococcal Enterotoxin N: Implications for Binding Properties to Its Cellular Proteins"

_ijms, 2019, doi:10.3390/ijms20235921_

Round 1

Reviewer 1 Report

This manuscript by Zeng et al., reports the first x-ray structure of Staphylococcal enterotoxin N (SEN). The authors conclude that several critical residues for receptor interaction is different in SEN compared to other SEs suggesting that SEN may adopt a different binding mode to its receptors.

The data presented is interesting and is clearly novel since there are no previous reports on the structure of SEN superantigen. However, the authors do not fully analyse the data, as it is written in a very descriptive format.

Specific comments:

To publish the data, the authors need to discuss the consequences of the residues that are differing from the other SAgs, will that affect the binding properties and specificity, and how? The authors conclude in the abstract that “SEN may adopt a different binding mode to its cellular receptors” this needs to be thoroughly discussed and suggestions should be clear on how the binding mode will be different or similar, and what are the specific consequences of that. MD simulations could potentially be used to confirm the results.  

The evolutionary classification of SAgs (including SEN into group III) was published already in 2007 (by Brouillard et al JMB) that should be referenced instead of claiming that the authors sequence alignment revealed that (page 3 line 85).

The nomenclature of TCRVβ was changed more than 15 years ago to the TRBV nomenclature, please amend the manuscript accordingly.

Author Response

Response to Reviewer 1 Comments

Point 1: To publish the data, the authors need to discuss the consequences of the residues that are differing from the other SAgs, will that affect the binding properties and specificity, and how? The authors conclude in the abstract that “SEN may adopt a different binding mode to its cellular receptors” this needs to be thoroughly discussed and suggestions should be clear on how the binding mode will be different or similar, and what are the specific consequences of that. MD simulations could potentially be used to confirm the results. 

Response 1: Response: Thanks for the profound comment. In this study, we tried to dig the structural characteristics of SEN, in particular of structural insight into cellular receptor binding (mainly TRBV, MHCα, and MHCβ) based on a X-ray structure alone. In this situation, it is difficult to identify the specific amino acid accounting for the receptor engagements. However, several pioneering crystallographic studies on SAg-receptor complexes provide significant information about their interactions. According to the conservations and divergences of those critical binding regions (or amino acid residues) in SEN structure, we proposes the receptor bindings properties of SEN which are different from the reported SAgs. Specifically, at the most critical TCR binding region, the only conservatively present amino acid in SEN is an asparagine, and the common existing aromatic pair does not exist. The MHCα binding site also has the significant amino acid variations, but the MHCβ site is conserved. As you suggestion, MD simulation is a proper tool to investigate the consequences of amino acid variations with regard to receptor bindings. However, the present condition of our research team does not allow to perform the MD simulation study. We used a simplified approach. First, a primary SEN-receptor complex was generated by structural superposition SEN with other structure complexes; second, energy minimization was performed to primary SEN-receptor complex. These modeled SEN-receptor complexes provide the specific intermolecular interactions and overall interface areas. These results have added to the revised manuscript. In addition, we wrote a conclusion section in the revised text, which summarizes the structural properties of SEN.

Point 2: The evolutionary classification of SAgs (including SEN into group III) was published already in 2007 (by Brouillard et al JMB) that should be referenced instead of claiming that the authors sequence alignment revealed that (page 3 line 85).

Response 2: Thanks for the suggestion. We did neglect the important study. We have quoted the work by Brouillard et al. in the text where proposes the evolutionary group of SEN.

Point 3: The nomenclature of TCRVβ was changed more than 15 years ago to the TRBV nomenclature, please amend the manuscript accordingly.

Response 3: Thanks for your suggestion. We should follow the current nomenclature of TCR Vβ domain. We have made change throughout the article. 

Reviewer 2 Report

Excellent work, enlightening the structure and biological properties of staphylococcal enterotoxin N.

Author Response

Response to Reviewer 2 Comments

Point 1: Excellent work, enlightening the structure and biological properties of staphylococcal enterotoxin N.

Response 1: Thank for your appreciation of our study.

Reviewer 3 Report

This study is complete and I am thanking you for doing good work.

What is the link for your protein crystal in PDB/rcsb.org?

Authors English is decent and scientific presenting skill is also very nice. However my few concerns are as follows and they are random:

Why author need PDB code: 1L05 as a molecular replacement of structure (Crystal structure of the D227A variant of Staphylococcal enterotoxin A). Please justify why your study is different from Peterrsson K et all study (https://www.ncbi.nlm.nih.gov/pubmed/?term=12467569). Please highlights the differences other than diffraction; such as his was 3.2 A and yours is 1.8A. Also include PDB: 1HXY, PDB: 2G9H for justification. I want to know; what is different from previously published 2 crystals with MHCII site?

I don’t see RMSD graph?

What is the chemical composition of Tacsimate (which you are successful in making crystal)?

I don’t see the use of IPTG in BL21 culture; pGEX-6p is IPTG inducible? Was GST there in plasmid or DNA insert? Can you put the picture of protein after GST cleavage and MONO Q separation and purified protein also?

Put the information of Amino Acid sequence of crystalized protein with the domains information. If possible label them nicely along with site of MHC interaction. If possible you can put immunogenicity index of stretch of sequences.

Author Response

Response to Reviewer 2 Comments

Point 1: What is the link for your protein crystal in PDB/rcsb.org?

Response 1: The atomic coordinate and structure factor of SEN structure presented in this research have been deposited in the Protein Data Bank (PDB) under accession entry 6KRY. The information has been indicated in the Material and Methods section (4.4. Data collection, processing, and structure determination) of manuscript.

Point 2: Why author need PDB code: 1L05 as a molecular replacement of structure (Crystal structure of the D227A variant of Staphylococcal enterotoxin A). Please justify why your study is different from Peterrsson K et all study (https://www.ncbi.nlm.nih.gov/pubmed/ ?term= 12467569). Please highlights the differences other than diffraction; such as his was 3.2 A and yours is 1.8A. Also include PDB: 1HXY, PDB: 2G9H for justification. I want to know; what is different from previously published 2 crystals with MHCII site?

Response 2: Thanks for the profound comments. In this study, we tried to dig the structural characteristics of SEN, in particular of structural insight into cellular receptor binding (mainly TRBV, MHCα, and MHCβ) based on a X-ray structure alone. In this situation, it is difficult to identify the specific amino acid accounting for the receptor engagements without those pioneering crystallographic studies on SAg-receptor complexes which provide significant information about their interactions. According to the conservations and divergences of those critical binding regions (or amino acid residues) in SEN structure, we proposes the receptor bindings properties of SEN.

1L05, 1HXY, and 2G9H are the structures of SEA D227A variant, SEH, and SEI in complex MHC molecules, respectively. SEA and SEH belong to evolutionary group III, and SEI is group V SAg. SAgs in both evolutionary groups III and V have two MHC binding sites. 1L05 structure represents the low-affinity MHC binding site (SAg binds to MHCα). 1HXY and 2G9H structures represent the high-affinity MHC binding site (SAg binds to MHCβ). SEN is a new member of group III SAg. In our study, we analyze the critical amino acids or regions involved in MHC binding that were proposed by the above three studies in SEN structure. The MHCα binding site on SEN has significant amino acid variations, in particular those on the hydrophobic ridge which protrudes into a hydrophobic pocket in MHCα. The MHCβ site in SEN is largely conserved.

Point 3: I don’t see RMSD graph?

Response 3: When we compared the overall structural divergences between SEN and other SAg structures, their α-carbon RMSD values were calculated. For your question, we have no idea to present many such kind of graph as a figure.

Point 4: What is the chemical composition of Tacsimate (which you are successful in making crystal)?

Response 4: Tacsimate is a mixture of titrated organic acid salts composed of 1.8305 M malonic acid, 0.25 M ammonium citrate tribasic, 0.12 M succinic acid, 0.3 M DL-malic acid, 0.4 M sodium acetate trihydrate, 0.5 M sodium formate, and 0.16 M ammonium tartrate dibasic. It was developed by Hampton Research Corp. and was used for protein crystallization (Protein Science, 10: 418-422, 2001). Tacsimate was used to prepare a number of crystallization conditions in crystallization kits of Hampton Research.

Point 5: I don’t see the use of IPTG in BL21 culture; pGEX-6p is IPTG inducible? Was GST there in plasmid or DNA insert? Can you put the picture of protein after GST cleavage and MONO Q separation and purified protein also?

Response 5: We mentioned use of isopropyl-β-D-thiogalactoside for induction. Because the compound only appear one time in the manuscript, the abbreviation, IPTG, was not given in the text. The pGEX expression vectors (GE Healthcare) contain GST coding sequence and are used to express GST fusion proteins. The recombinant proteins produced by pGEX vectors contain GST tag at their N-terminal with a PreScission protease site between GST and target protein. The transcription of GST and target protein is controlled by tac promoter (a hybrid between the trp and lac UV5 promoters), therefore the recombinant protein production can be induced using IPTG.

In the revised manuscript, we prepared a supplemental figure (Figure S1) which presents results of SDS-PAGE analysis throughout recombinant SEN production and purification. In the Figure, the whole cell lysate after IPTG induction, SEN after GST affinity purification, GST tag cleavage, MONO Q, and gel filtration are shown.  

Point 6: Put the information of Amino Acid sequence of crystalized protein with the domains information. If possible label them nicely along with site of MHC interaction. If possible you can put immunogenicity index of stretch of sequences.

Response 6: We modified Figure 2. In the new figure, the domains information and the amino acids involved in receptor binding are indicated. Honestly, we do not get your idea to show the immunogenicity index with sequence. In the graphical abstract of the manuscript, all the functional regions (TRBV binding site, MHCα binding site, MHCβ binding site, and CD28 binding site) which interact with immune receptors are presented in different colours on SEN surface.

Round 2

Reviewer 1 Report

The authors have improved the manuscript, but they seem not to be understanding the difference between the new and the old TCR nomenclature (check IMGT data base).  You cannot simply change the name TCRVβ to TRBV, the numbers is also changing upon nomenclature change. This must be corrected.  

Author Response

Response to Reviewer 1 Comments

Point 1: The authors have improved the manuscript, but they seem not to be understanding the difference between the new and the old TCR nomenclature (check IMGT data base).  You cannot simply change the name TCRVβ to TRBV, the numbers is also changing upon nomenclature change. This must be corrected. 

Response 1: Thanks for the comment. We have carefully checked the TCR nomenclature in the manuscript and learned the useful immunogenetic database that you recommended to us. In the IMGT database, for human, there are 30 IMGT TRBV subgroups (TRBV1, TRBV2…TRBV30). In most subgroups, there are several IMGT genes. For example, subgroup TRBV7 has 9 IMGT genes: TRBV7-1, TRBV7-2…TRBV7-9. Those TRBV subgroups containing only one gene, the gene and subgroup nomenclatures are same. It seems still unclear to us with regard to the TRBV nomenclature because the usages of nomenclature of TRBV gene and nomenclature of TRBV subgroup are confusing in the literature. In our manuscript, we use hTRBV7-9 representing human TCR bearing TRBV7-9 (reference 18, Sci. Rep. 2016, PDB entries: 5FK9, 5FKA; reference 22, PDB entry: 4C56). We keep the nomenclature  consistent with the original literature and PDB entries. If we still do not understand the rule for present TCR nomenclature, we are very willing to amend them according to your instruction.

Reviewer 3 Report

The current version of paper is much improved and I thanks authors for submitting crystal data in RCSB.org. Hope it will be available to public access soon.

Authors confused about my RSMD Question; since, you have mentioned ‘…..RMSD values between the 267 structures were calculated using SuperPose (http://superpose.wishartlab.com/)......’ I wanted authors to put simulation trajectory in supplementary.

Best

Author Response

Response to Reviewer 3 Comments

Point 1: Authors confused about my RSMD Question; since, you have mentioned ‘…..RMSD values between the structures were calculated using SuperPose (http://superpose.wishartlab.com/)......’ I wanted authors to put simulation trajectory in supplementary.

Response 1: In molecular dynamics, protein structure changes over time are compared to the starting structure. In this situation, RMSD values are calculated for all frames in the trajectory, and a plot of RMSD vs. time can be generated. Monitoring the RMSD values can give insights into its structural conformation throughout the simulation. In our study, molecular dynamics was not performed. The numerical RMSD values in the manuscript are only used to represent the difference between a target structure (SEN) and each reference X-ray structure (other SAgs). The usage of RMSD are usually seen in the structural studies. For example, Rödström et al. used RMSD values to assess three-dimensional similarities between SEA and other SAgs (reference 18, Sci. Rep. 2016), and Saline et al. used  RMSD value for comparing different SEH-TCR complexes (reference 17, Nat. Commun. 2010).

We think that the related description (RMSD values between the structures were calculated using SuperPose) in the Material and Methods in not clearly presented. We modified the statement about RMSD calculation to “Pairwise RMSD values between SEN and other SAg structures were calculated using SuperPose”. In addition, we prepared a new table (Table S1) which summarises the information of each reference SAg structure (PDB entries, chain ID used for calculation, and amino acid numbers in the chains) and each pairwise RMSD with SEN.